# Genome-Wide Identification and Transcriptional Expression of the *METTL21C* Gene Family in Chicken

**DOI:** 10.3390/genes10080628

**Published:** 2019-08-20

**Authors:** Ge Yang, Hongzhao Lu, Ling Wang, Jiarong Zhao, Wenxian Zeng, Tao Zhang

**Affiliations:** School of Biological Science and Engineering, Shaanxi University of Technology, Hanzhong, Shaanxi 723001, China

**Keywords:** chicken, *METTL21C*, evolution, expression profiling

## Abstract

The chicken is a common type of poultry that is economically important both for its medicinal and nutritional values. Previous studies have found that free-range chickens have more skeletal muscle mass. The methyltransferase-like 21C gene (*METTL21C)* plays an important role in muscle development; however, there have been few reports on the role of *METTL21C* in chickens. In this study, we performed a genome-wide identification of chicken *METTL21C* genes and analyzed their phylogeny, transcriptional expression profile, and real-time quantitative polymerase chain reaction (qPCR). We identified 10 *GgMETTL21C* genes from chickens, 11 from mice, and 32 from humans, and these genes were divided into six groups, which showed a large amount of variation among these three species. A total of 15 motifs were detected in *METTL21C* genes, and the intron phase of the gene structure showed that the *METTL21C* gene family was conservative in evolution. Further, both the transcript data and qPCR showed that a single gene’s (*GgMETTL21C3)* expression level increased with the muscle development of chickens, indicating that the *METTL21C* genes are involved in the development of chicken muscles. Our results provide some reference value for the subsequent study of the function of *METTL21C.*

## 1. Introduction

The chicken (*Gallus gallus domesticus*) may originate from red junglefowl (*G. gallus*) [1]. Lueyang black-bone chicken is a famous native chicken breed with some distinctive features, such as its low fat content, strong muscles, and considerable nutritional and medicinal values. It has been listed in the Poultry Genetic Resources of China and was identified as a National Geographical Indication Product [2]. Chicken feeding methods are generally divided into caged and free range. It has been found that free-range chickens have more muscle mass [3]. A previous study showed that METTL21C is a skeletal-muscle-specific lysine methyltransferase [4]. It is well known that skeletal muscle plays an important role in the development of chickens, and can affect the athletic ability of the chickens. Thus, it is relevant to explore the role of the *METTL21C* gene in the muscle development of chicken species.

The methyltransferase-like 21C gene (*METTL21C*) belongs to the *METTL21* family of the methyltransferase superfamily; in humans, *METTL21C* is located on 13q33.1 and is also called C13orf39 [5]. METTL21C, which has protein-lysine N-methyltransferase activity and participates in protein modification [6], is one of the protein methyltransferases (MTases) and exists in most eukaryotic organisms. It catalyzes the transfer of a methyl group from the cofactor S-adenosylmethionine (AdoMet) to a substrate and mainly occurs on lysine and arginine. MTases catalyze different biological reactions in organisms, which regulate many different substrates, such as lipids, RNA, DNA, small molecules, and proteins [7,8,9]. METTL21C, which performs regular protein structure housekeeping such as folding and proofreading, is also a 70 kDa heat shock protein which acts as an ATP-dependent (Adenosine triphosphate dependent) molecular chaperone [10]. Further, bivariate genome-wide association studies (GWAS) have found that METTL21C can also act as a pleiotropic factor on muscles [11]. METTL21C can regulate calcium homeostasis in bones and muscles and promote the differentiation of myoblasts to myotubes through the nuclear factor (NF)-kB signaling pathway [11]. In addition, METTL21C may also reduce apoptotic osteocytes and regulate intracellular homeostasis [12]. A previous study concluded that METTL21C is an important regulator of protein degradation in skeletal muscles under normal and enhanced proteolytic conditions [4]. This suggests that *METTL21C* is closely related to the development of muscles and thus deserves further exploration.

The paralog of METTL21C is METTL21D, which is also a protein methyltransferase. Widely found in eukaryotes, METTL21D can catalyze the trimethylation of Lys315 in valosin-containing protein (VCP) in vitro, so it is also called VCP-KMT (valosin containing protein lysine methyltransferase) [6]. VCP is found in all eukaryotes and is a highly conserved AAA+ protein (ATPase associated with various cellular activities) [13]. It is a homotyped hexamer cyclic molecule comprising four domains: a flexible N-terminal domain and a short C-terminal domain, which may bind to the substrate, and two AAA ATP-binding domains [14,15,16]. VCP participates in a wide range of biological activities, including homotypic membrane fusion [17], endoplasmic reticulum-related degradation [18], DNA repair [19], and mitochondrial autophagy [20]. VCP and hereditary inclusion body myopathy are associated with Paget disease and frontotemporal dementia (inclusion body myopathy with early-onset Paget disease and frontotemporal dementia (IBMPFD)), which primarily affects muscles and the brain [21]. The pathogenesis of IBMPFD is due to the autosomal dominant single amino acid replacing the VCP terminal domain and the highly conserved residues within the D1 domain [21,22]. As mentioned above, VCP includes four domains; the domain D1 is one of the two AAA ATP-binding domains and so the mutation of the *VCP* gene is involved in the pathogenesis of IBMPFD [21]. In a recent study, it was found that METTL21D could promote tumor metastasis and affect cell growth, migration, and infection [23]. Since METTL21C is a homologous sequence of METTL21D, it was found that METTL21C interacts with VCP and catalyzes the modification and trimethylation of lysine 315 [4]. The *METTL21C* gene family from *G. gallus domesticus* offers an exceptional opportunity to study the evolution and function of muscle development systems [24]. In this study, to detect the mechanism of *METTL21C* in the development of chicken muscles, according the global genomic data on chickens, mice, and humans published by others, we systematically characterized 10 *METTL21C* genes of *G. gallus*, 11 *METTL21C* genes of *Mus musculus*, and 32 METTL21C genes of *Homo sapiens*. We used an iterative process of manual and computational analyses to identify examples of *METTL21C* in *G. gallus*, *M. musculus,* and *H. sapiens* encoding a gene family within the latest-released *G. gallus*, *M. musculus,* and *H. sapiens* whole-genome sequences [25,26,27]. We constructed a phylogenetic tree of the *METTL21C* gene family, the conserved motifs, gene structure, protein domain, and chromosome localization and further analyzed *G. gallus METTL21C* genes. To understand the function of the *METTL21C* genes in *G. gallus*, we also studied the transcriptional-level expression profiles of *METTL21C* of different tissues at different developmental stages. Our results provide useful theoretical support for the functional characterization of these *METTL21C* genes that are involved in the age-development process of *G. gallus*.

## 2. Materials and Methods

### 2.1. Identification of METTL21C in Sequenced Genomes of G. gallus

The query sequences of the *METTL21C* genes downloaded from the National Center for Biotechnology (NCBI) database (http://asia.ensembl.org/info/data/ftp/index.html) were for *G. gallus*, *M. musculus*, and *H. sapiens*. The whole protein sequence of *G. gallus* (http://asia.ensembl.org/info/data/ftp/index.html?tdsourcetag=s_pcqq_aiomsg) was also downloaded from the NCBI database (http://asia.ensembl.org/info/data/ftp/index.html). Based on these sequences, using the BLASTP (basic local alignment search tool) software with E-value parameters less than 1e-10, a total of 22 genes were examined as the candidate sequence. To further check the members of the *METTL21C* gene family, we used a profile hidden Markov model (HMM) implemented with default parameters in HMMER v3.2.1 for Windows (http://hmmer.org/download.html/) according to the domain named pfam10007 [28].

### 2.2. METTL21C Protein Alignment, Phylogenetic Analysis, Pfam Domain Detection, and Chromosome Location Analysis of G. gallus METTL21C Genes

MEGA software was used to do a complete alignment based on the checked sequence with default settings [29], and an unrooted phylogenetic tree was constructed based on alignments using the neighbor-joining (NJ) method with pairwise deletion of 1000 bootstraps and a Poisson model. According to this phylogenetic analysis result, we classified these genes into different groups [30,31], and the conserved domains of these 119 *METTL21C* protein sequences were detected in Batch software (https://www.ncbi.nlm.nih.gov/Structure/bwrpsb/bwrpsb.cgi). Those domains found by Pfam were also detected by the SMART program (http://smart.embl-heidelberg.de/) with an E-value cutoff of 1.0 to validate the final result. Based on the GFF (general feature format) file, we anchored these genes to the related genomes through the MapChart software (http://mapchart.software.informer.com/2.2/) using the default parameters. The GFF file of these three species was downloaded from the NCBI database (https://www.ncbi.nlm.nih.gov/genome/?term=).

### 2.3. Gene Structure and Motif Analysis of METTL21C Genes

The exon–intron structure was illustrated using the online gene structure display server program (http://gsds.cbi.pku.edu.cn/) according to the information in the GFF file of the *METTL21C* gene family. The motifs were identified using the multiple EM (emphasize) for motif elicitation (MEME) program using the default parameters [32]. The parameters were as follows: the maximum number of motifs was set to 20, and the optimum motif width was set to 30–50.

### 2.4. The Materials, Treatments, and Collections of Different Breeds of Chickens

A total of six different breeds of black-bone chickens were collected, they included 20-day-old Liangfenghua chickens, 20-day-old Qingjiaoma chickens, 60- and 120-day-old Siyu chickens, and 60- and 120-day-old Lueyang black-bone chickens. For each chicken development time, we considered periods of 20, 60, and 120 days. We raised these chickens in cages or free-range: Lueyang black-bone chickens were free-range and the rest were caged. Different breeds of chickens had different feeding conditions. All the chickens are under normal circumstances, the optimal temperature for each stage of silky fowl through the brooding period (0~6 weeks) is as follows: 0~3 days is 38~36 °C, with a decrease of 2~3 °C every week to 6 weeks. The temperature should be gradually reduced by 21~18 °C; the optimum temperature for broiler chickens after 6 weeks of age is 21~16 °C. The humidity in the house should be controlled by a hygrometer. The optimum suitable humidity for chicks is 55%~65%, and that of adult chickens is 60%~70%. Regarding the illumination, chicks of 1 to 3 days old maintain 23 to 24 h of light, 4 to 14 days of age maintain 16 to 19 h of light, and after 15 days of age, 8 to 9 h of light are maintained, and artificial light is applied when sunshine is insufficient. Free-range care generally does not require artificial control of the growing environment, and if necessary, some measures are taken according to the weather.

### 2.5. Materials, Treatments, and Collection of the Transcriptome Profile of the METTL21C Gene Gamily of Chickens

We put the samples in liquid nitrogen prior to storage at −80 °C until use [33]. Total RNA was isolated using an RNA-prep Pure Kit (Tiangen, China) [34,35]. Libraries for RNA sequencing (RNA-seq) were produced using NEBNext Ultra RNA Library Prep Kit (NEB, Beverly, MA, USA). Paired-end sequencing was performed on the Illumina HiSeq2500 platform to generate 100 bp reads using default parameters (Novogene Bioinformatics Technology Co., Ltd., Beijing, China (www.novogene.cn)). The de novo transcriptome was assembled using default settings in Trinity [36] and was based on the well-genome reference of chickens [37]. The assembled RNA-seq data were used to quantify these gene expression levels based on their fragments per kilobase of exon per million reads mapped (FPKM) values using Cufflinks with default parameters [38]. For each member’s expression level of the *METTL21C* gene family, we used the HemI 1.0 software with default parameters [32,39]. For the expression level of several *METT21C* genes with error bars, the ggplot2 R package was used [40].

### 2.6. Quantitative Real-Time PCR

Because METTL21C is related to muscle development, we collected the skeletal muscle of different breeds at the same anatomic sites, such as the soleus muscle for the Siyu, Lueyang, Liangfenghua, and Qingjiaoma chickens. We also collected stomach, gizzard, lung, skeletal muscle, liver, skin, and myocardium from the Lueyang chickens (Appendix A). Total RNA was extracted from individual chicken tissues (stomach, gizzard, lung, skeletal muscle, liver, skin, and myocardium) and reverse transcribed to produce first-strand cDNA as described above. Real-time amplification reactions were performed on a 7500 fast real-time PCR system (Applied Biosystems, USA). The relative expression level of each gene was calculated according to the 2*^−ΔΔCt^* method [41]. The *β-actin* gene was used as an internal control [42], and all analyses were performed with three technical and three biological replicates (Appendix A).

## 3. Results

### 3.1. Identification and Phylogenetic Analysis of METTL21C Genes

To identify *METTL21C*-encoding genes, we seeded files for the lysine methyltransferase domains from *G. gallus*, *M. musculus*, and *H. sapiens*. We acquired 119 candidate *METTL21C* genes from these three species, including 22 *G. gallus* genes, 31 *M. musculus* genes, and 66 *H. sapiens* genes. To check for the presence of the lysine methyltransferase domain, those 119 candidate protein sequences were reduced to 53 genes that contain the same conserved domain as the lysine methyltransferase. Finally, we found 53 *METTL21C* genes in all species. In order to explore the evolutionary relationship among the members of the *METTL21C* families, an NJ tree based on *G. gallus*, *M. musculus*, and *H. sapiens* protein sequences was constructed (Figure 1). These three species had 10, 11, and 32 *METTL21C* genes, respectively. Based on the completed alignment of the sequences and the phylogenetic analysis, we divided a total of 53 members of chickens, mice, and humans into six groups, named groups I to VI. (Appendix A). We found that group I contained 8 *METTL21Cs*, group II contained 10 *METTL21Cs*, group III contained 6 *METTL21Cs*, group IV contained 6 *METTL21Cs*, group V contained 5 *METTL21Cs*, and group VI contained 18 *METTL21Cs*. We found four *G. gallus METTL21Cs* (*GgMETTL21C7, GgMETTL21C8, GgMETTL21C4*, and *GgMETTL21C3*) to be closely associated with *M. musculus*, and six *G. gallus METTL21Cs* (*GgMETTL21C9*, *GgMETTL21C5*, *GgMETTL21C1*, *GgMETTL21C2*, *GgMETTL21C6*, and *GgMETTL21C10*) were found to be closely associated with *H. sapiens*.

### 3.2. Chromosomal Locations of METTL21C Genes and Their Relationship with G. gallus

The 10 *METTL21C* genes were distributed unevenly in the *G. gallus* chromosomes based on the physical positions. We found the distribution of the *METTL21C* genes were different on each of the six chromosomes (Table 1; Figure 2). The accession number of the coding sequence and the genomic and protein IDs are listed in Table 1. Chromosomes 8 and 18 contained only one gene, *GgMETTL21C1* and *GgMETTL21C6*, respectively, whereas *GgMETTL21C3* and *GgMETTL21C4* were both found on chromosome 1, but in different positions. *GgMETTL21C2* and *GgMETTL21C5*, *GgMETTL21C7* and *GgMETTL21C8*, as well as *METTL21C9* and *GgMETTL21C10* were found at different positions chromosomes chr14, chr7, and chr5, respectively (Figure 2).

### 3.3. METTL21C Gene Structures and Conserved Motifs

To better understand the relationship between the structure and functions among *METTL21C* genes of *G. gallus*, *M. musculus*, and *H. sapiens*, the exon/intron organization and conserved motifs were analyzed and are shown in Figure 3. To compare the structural components of the 53 *METTL21C* genes, their structures, including exons and introns, were mapped. The maximum number of exons was found in *MmMETTL21C9* (nine exons), and the minimum number of one exon was found in *GgMETTL21C6*. It seems that the most closely related members had similar numbers of exons. For example, those in Group I mostly had one exon, such as *GgMETTL21C6*, *HsMETTL21C1*, *HsMETTL21C2*, *MmMETTL21C3,* and *MmMETTL21C4*, but *HsMETTL21C23* had six exons and *GgMETTL21C10* had five exons. Members in Group II mostly had three exons, Group III mostly had nine exons, Group IV mostly had three exons, Group V had a varying number of exons (three to five), and Group VI mostly had three exons. Interestingly, we found there were one to nine exons in *G. gallus*, for example, *GgMETTL21C6* had one exon, while *GgMETTL21C5* had nine exons.

We also found using the MEME program that the *METTL21C* genes have 15 motifs. Motif 1 was mostly present in all breeds except *GgMETTL21C7*, *HsMETTL21C7*, *HsMETTL21C14*, *HsMETTL21C22*, and *GgMETTL21C10*. Motif 2 appeared in *G. gallus*, except *GgMETTL21C9*. As expected, the most closely related members in the same subfamilies shared a common motif composition, which may be indicative of similar function. These results suggest that all *METTL21C* genes in *G. gallus* share at least one typical domain.

### 3.4. Expression Profile Analysis of G. gallus METTL21C Genes

To understand the expression patterns of different genes in the *METTL21C* gene family, we tested the expression levels of 10 genes in six samples from four breeds of chickens based on RNA-seq data (Appendix A). Based on phylogenetic analysis and expression levels, we divided the 10 members of chickens into three groups, named groups A, B, and C (Figure 4). Meanwhile, based on the different expression levels, we noticed that the expression levels of GgMETTL21C4, *GgMETTL21C6*, *GgMETTL21C5*, and *GgMETTL21C10* were lowest and there was no difference in the different breeds and different development periods. Our results showed that *GgMETTL21C7* and *GgMETTL21C8* had relatively high expression in *G. gallus*, and there were some differences among the different samples. The expression patterns of *GgMETTL21C7* and *GgMETTL21C8* have breed specificity. Interestingly, we also detected that *GgMETTL21C7*, *GgMETTL21C8*, and *GgMETTL21C2* genes had lower expression of Ly120 compared with Ly60. However, we found that *GgMETTL21C3* had a large difference between chickens of different varieties. *GgMETTL21C3* was highly expressed in Lueyang chickens and had a higher expression of Ly120 compared with Ly60.

### 3.5. Real-Time PCR of METTL21C Genes of the METTL21C Gene Family

These different breeds of chickens were not significantly different for growth in morphology, based on quantitative real-time PCR analysis of the expression of *METTL21C3* in Sy and Ly chickens. The results showed that *METTL21C3* was highly expressed in 120 days of Ly chickens (Figure 4 and Figure 5a). Using β-actin as the internal reference gene, qPCR was used to detect the expression level of *METTL21C3* in seven chicken tissues of Ly chickens, and it was found that the expression levels of myocardial and skeletal muscles were higher (Figure 5c). Based on quantitative real-time PCR analysis of the expression of *METTL21C3* in soleus muscles and toe elongation muscles of Qjm and Ly chicken, we found that the expression level of *METTL21C3* increased significantly with age (Figure 5d,e). Finally, we found that the expression level of *METTL21C3* in free-range chickens was higher by comparing the expression levels of *METTL21C3* in caged chickens and free-range chickens, further indicating that the *METTL21C3* gene is indeed associated with muscle growth and development.

## 4. Discussion

### 4.1. The Characteristics of the METTL21C Gene Family

*METTL21C* is a skeletal-muscle-specific lysine methyltransferase which plays a critical role in the development of bones and muscles [4]. The *METTL21C* gene family is closely related to osteoporosis and sarcopenia by the NF-kB signaling pathway to prevent the differentiation of myoblasts [11]. The *METTL21C* genes of chickens contained three conserved domains identified in NCBI-Batch-CDD (NCBI Batch conserved domain database), including the AdoMet_MTases superfamily, SAM_superfamily, and FAM86, suggesting that *METTL21Cs* are extremely conservative in evolution (Appendix A). We found all chicken *METTL21C* genes had the conserved domain of AdoMet_MTases. The key domain is AdoMet_MTases (Appendix A), and in general, the most common methyl donor is S-adenosyl-L-methionine (AdoMet). The methylation reactions often occur in cells, which are catalyzed by MTases and induce methyl transfer from donor substrate molecules, implying that these genes have a similar function (Appendix A) [43]. In chickens, the number of domains of AdoMet_MTases is larger than that of other domains, such as the domain of SAM and FAM86.

The gene family is structurally and functionally similar and contains a common domain. For the *METTL21C* gene, the lysine methyltransferase domain was determined in a total of 10 member groupings in the *METTL21C* gene family. Based on phylogenetic analysis, the *METTL21Cs* are divided into six subfamilies named Groups I to VI. The present study identified 10 *METTL21C* genes in chickens, and more in *M. musculus* and *H. sapiens* (Table 1). The proteomes of these animals are different. For example, the *M. musculus* proteome is about 46,930 kb, and the *H. sapiens* proteome is about 70,940 kb. However, the *G. gallus* proteome is about 25,209 kb. Comparative analysis suggested that chickens, mice, and humans have similar numbers of gene families [26,27,44], indicating that the number of *METTL21Cs* in different species is not determined by the genome size (Table 1).

In previous studies, *GgMETTLC1* was called *METTL23*, which is a GABPA (GA-binding protein transcription factor, α subunit) function regulator that plays an important role in the transcriptional regulation of human cognition [45,46] and affects the nervous system and brain development [47]. Thus, we speculated that *GgMETTL21C1* could be related to brain development in chickens. In terms of *GgMETTL21C2*, it contains the FAM86A protein that belongs to the protein MTases family, which catalyzes the trimethylation of eukaryotic elongation factor 2 (eEF2) on Lys-525 [48], indicating that the *GgMETTL21C2* is involved in protein synthesis, which is critical for cell survival during stress [49]. In this study, *GgMETTL21C3* was highly expressed in Lueyang chickens, compared to other breeds (Figure 4 and Figure 5), and highly expressed in skeletal muscles compared to other muscles (Figure 5e). With the development of muscles in chickens, the expression level of *GgMETTL21C3* is increased, these results show that *GgMETTL21C3* is related to the development of skeletal muscles of Lueyang breeds (Figure 4 and Figure 5a–d). In a previous study, Philippe referred to *GgMETTL21C5* as *METTL22*, which can mediate methylation of kin17 on lysine 135. The kin17 is a DNA/RNA-binding protein that participates in the replication and repair of DNA and the process of pre-mRNA [50], and we found that *GgMETTL21C5* plays an important role in chickens. In a previous study, Magnus found that METTL21A acts as a protein MTase, which is involved in the trimethylation of a conserved lysiner [51], this study also named the unnamed ID, which will enable others to identify the gene more clearly (Appendix A).

### 4.2. The Evolution History of the METTL21C Gene Family of Chickens, Humans, and Mice

Gene structure is an important factor for gene family evolution [52]. The location of the gene coding sequence and the structure of the protein are largely determined by the position of the intron, and protein diversity is further increased by exon shuffling and alternative splicing [53,54]. The association of the intron phase with conservation at splice site sequences is related to the evolution of spliceosomal introns. Intron phase 0 indicates the highest conservation, intron phase 1 also shows high conservation, and intron phase 2 shows the lowest conservation [55,56]. In the case of the *METTL21C* genes in chickens, almost all of the genes’ structure was intron phase 0, so the *METTL21C* genes were conservative in evolution (Figure 3c). In conclusion, based on these analyses, including motif and gene structure, we showed that *METTL21C* genes were conservative in evolution. We found that the *METTL21C* gene structure of chickens is different from that of humans and mice. The structure of *METTL21Cs* in chickens has undergone evolutionary changes (Figure 2c).

Conserved motif analysis revealed that almost all the METTL21C proteins contained one to five subdomains (Appendix A). In addition, each subfamily had other special motifs, while some subfamilies also harbored specific motifs (Figure 3c). Interestingly, we found that the *METTL21C* genes of the same subfamily have a similar gene structure. This indicated that the gene structures are relatively conserved within each subfamily. For each subfamily, gene structures of *METTL21C* genes from chickens, mice, and humans are almost identical, suggesting that exon–intron structures of *METTL21C* genes are relatively conserved between these species. In particular, the *GgMETTL21C10* gene has a long genetic distance, a simple genetic structure, and only one domain, probably because of the lack of structure during the evolution process (Figure 3). Its domain may be the AdoMet_MTases superfamily. Motif 1 existed in almost the entire *METTL21C* gene family of these three species, which means that the *METTL21C* gene family is conservative in evolution (Figure 3a).

### 4.3. The Function of the METTL21C Gene Family of Chickens

To detect the expression level of the *METTL21C* gene family members in different breeds of chickens, we sequenced six RNA-seq from different species at different periods. We then examined the expression level of *METTL21C* genes in different species of chickens (Appendix A). Based on the phylogenetic analysis and *METTL21Cs*’ expression level, we divided these 10 genes into three groups. Group A included *GgMETTL21C3*, *GgMETTL21C4*, *GgMETTL21C9*, *GgMETTL21C7*, and *GgMETTL21C8*. Group B included *GgMETTL21C1*, *GgMETTL21C2*, and *GgMETTL21C6*. Group C included *GgMETTL21C5* and *GgMETTL21C10* (Figure 1). In previous studies, the *METTL21C* genes showed different expression patterns during different periods (20, 60, and 120 days) in Lfh, Qjm, Sy, and Ly (Appendix A). However, there have been few reports on the *METTL21C* gene family in chickens. As the number of days increased, we found that the expression of *METTL21C* also increased, implying that these genes of *METTL21C* play a critical role in the development of chickens (Figure 4). The results showed that the *METTL21C* gene family can catalyze the development of bone and muscle [8]. Interestingly, *GgMETTL21C7* and *GgMETTL21C8* are in the same subfamily (Figure 1) and their expression was almost the same, indicating that these two genes are related to the growth and development of chickens [4]. Group C had a low expression level (Figure 4). The expression level of *METTL21C3* increased with the growth and development of chickens, further indicating that this gene may inhibit apoptosis and promote bone and muscle development (Figure 4) [4]. Importantly, the transcript data showed that, compared with Ly60, *GgMETTL21C3* was highly expressed in Ly120, implying that *GgMETTL21C3* plays an important role in the development of Lueyang species. Otherwise, by comparing the relative expression levels of *METTL21C* in Sy and Ly at 60 days and 120 days, it was found that *GgMETTL21C* was highly expressed in Ly at 120 days (Figure 5a); this evidence showed that *GgMETTL21C3* is involved in the development of chickens. Therefore, we detected the relative expression of *Gg*METTL21C3 in different tissues of Ly chickens and found that the expression level was higher in skeletal muscle and myocardium (Figure 5b), further verifying that *GgMETTL21C* is related to skeletal muscle [4]. We also found that METTL21C is related to the growth and development of muscles in different breeds of chickens (Figure 5a,b) and in different feeding methods (Figure 5e). Both the transcript and qPCR analyses showed that *GgMETTL21C3* is involved in the muscle development of Lueyang black-bone chickens, which is consistent with previous studies [4,11,12].

## 5. Conclusions

In this study, we identified 10 *METT21C* genes in *G. gallus*. Phylogenetic analysis showed that the *METTL21C* genes can be divided into six groups based on specific and special domains. The 10 *GgMETTL21Cs* were respectively distributed on seven different chromosomes, with some colocalized to the same chromosome, such as *GgMETTL21C7* and *GgMETTL21C8* located in chromosome 7. Motif 1 was found in humans, mice, and chickens; Motifs 1, 5, 2, and 6 were found in most human *HsMETTL21C* genes; Motifs 1, 2, 5, and 8 were found in most mouse *MuMETTL21C* genes; and Motifs 1, 2, and 8 were found in most chicken *GgMETTL21C* genes. Based on this, we speculate that the conserved sequences are present in the *METTL21C* family. Expression profile analysis revealed that *METTL21C* gene family members display diverse expression patterns at different ages of different chickens. A total of three genes (*GgMETTL21C3 GgMETTL21C7*, and *GgMETTL21C8)* had high expression levels in *G. gallus*, suggesting they might play an important role in the development of chickens. In particular, the transcript data and qPCR analysis both showed that a single gene called *METTL21C3* is involved in the development of different muscles of the Lueyang species, such as the soleus, long extensor digitorum, myocardium, and leg muscles. This study provides strong evidence that *METTL21C3* may play a critical role in the development of chicken muscles.

## Figures and Tables

**Figure 1 genes-10-00628-f001:**
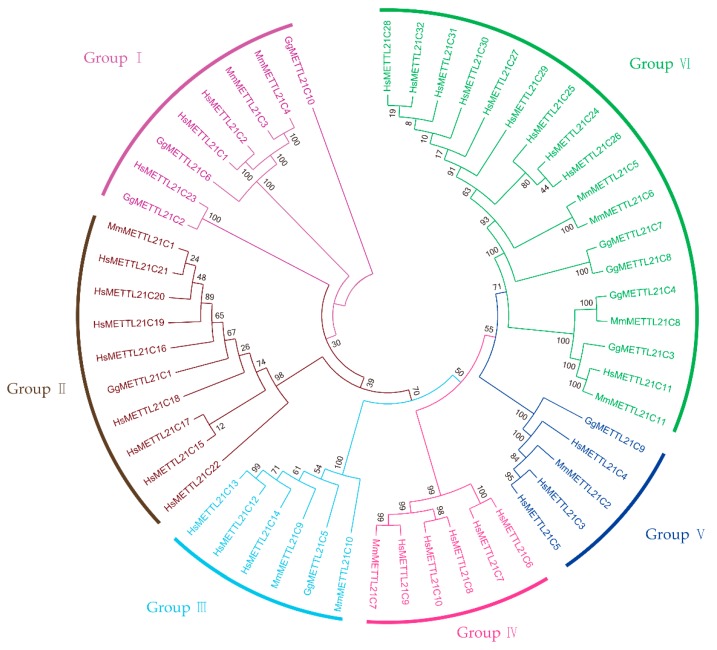
Phylogenetic analysis of methyltransferase-like 21C (METTL21C) proteins in *G. gallus domesticus*, *M. musculus*, and *H. sapiens*. These 53 sequences were used to construct a neighbor-joining (NJ) tree. The tree was divided into six groups (I–VI); the names of the different groups are displayed.

**Figure 2 genes-10-00628-f002:**
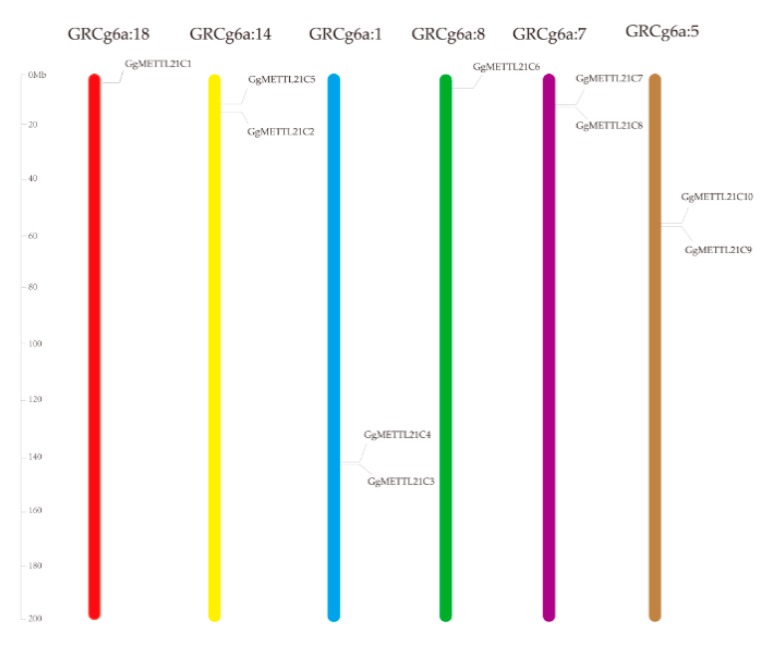
Chromosomal location in chickens of *METTL21C* genes on six chromosomes. Chromosomal locations in *G*. *gallus* are at the top of each bar. The scale of the chromosome is millions of base pairs (Mb).

**Figure 3 genes-10-00628-f003:**
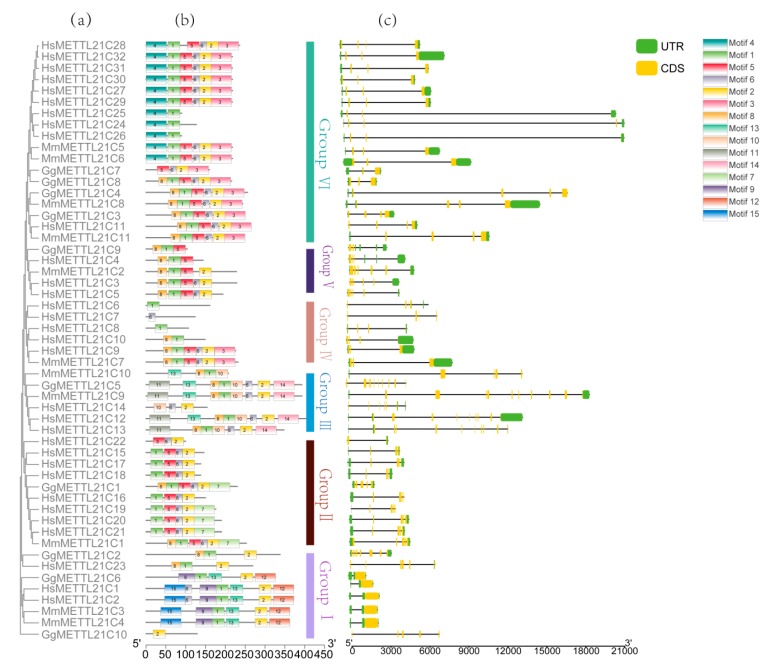
(**a**) Phylogenetic relationships, (**b**) motif compositions, and (**c**) gene structures of the 53 *METTL21C* genes identified in *G. gallus* (Gg), *M. musculus* (Mm), and *H. sapiens* (Hs). (**a**) Phylogenetic relationships using the NJ method, and different colors represent different groups. (**b**) Colored boxes indicate conserved motifs, and gray lines represent nonconserved sequences. The length of motifs in each protein is shown proportionally. (**c**) UTR represents upstream, CDS represents coding sequence, and EXON represents exon.

**Figure 4 genes-10-00628-f004:**
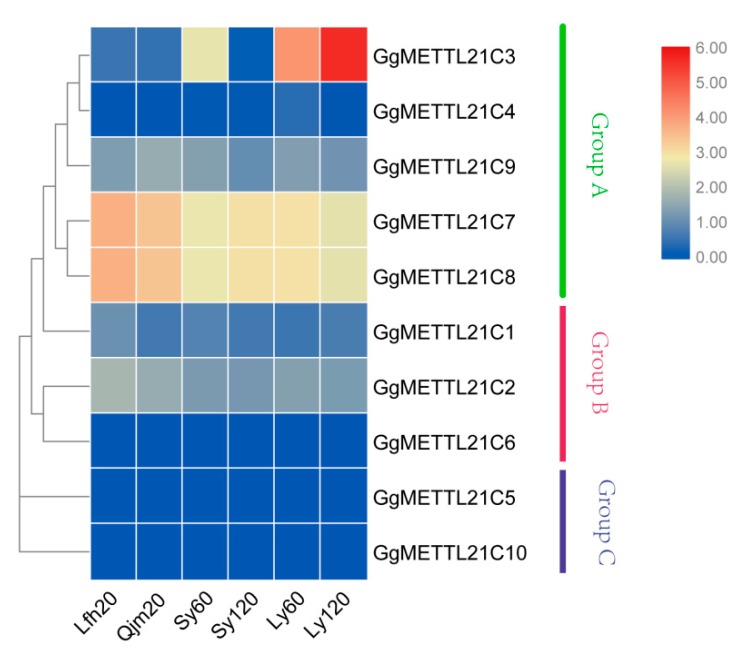
Expression patterns in *METTL21C* of *G. gallus.* The heat map was drawn in log10-transformed expression values. Red and blue represent relatively high and low expression, respectively, compared to the control. Lfh represents the chicken of Liangfenghua; Qjm represents the chicken of Qingjiaoma; Sy represents the chicken of Siyu; Ly represents the chicken of Lueyang. The numbers represent the age of the chickens in days.

**Figure 5 genes-10-00628-f005:**
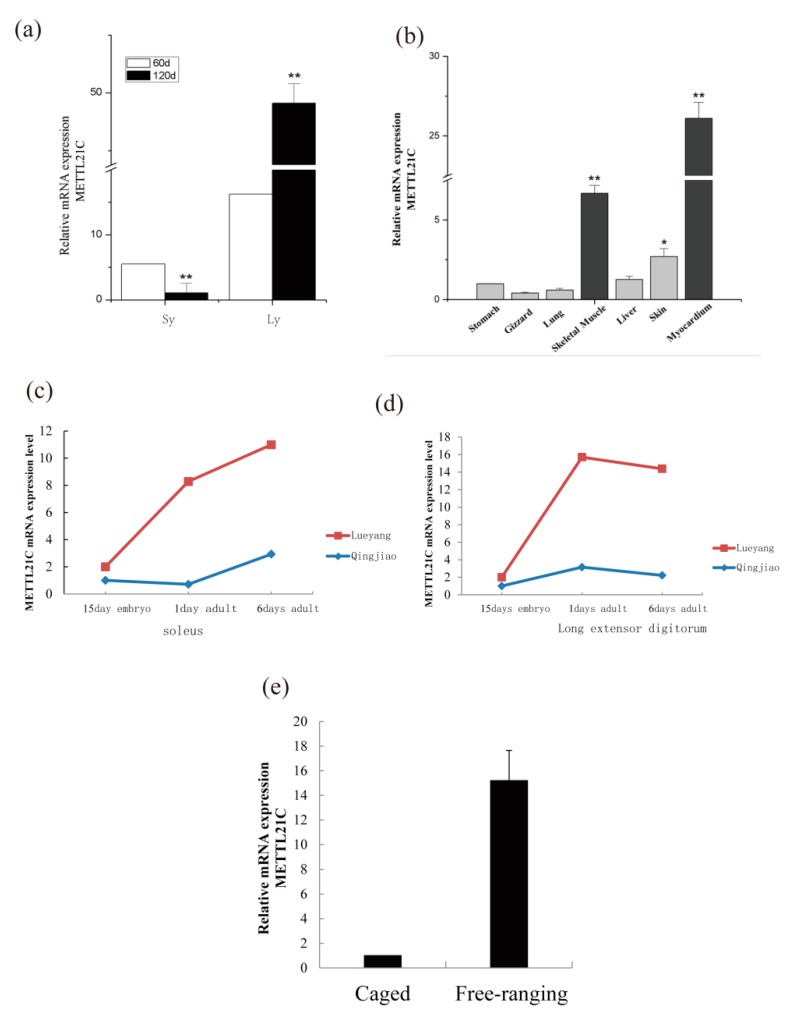
Relative mRNA expression: (**a**) *METTL21C* relative mRNA expression expression in different breeds of Sy and Ly chicken. (**b**) *METTL21C* relative mRNA expression in different tissues of Ly chicken. (**c**) *METTL21C* mRNA expression level in the soleus muscle. Lueyang represents Lueyang chickens and Qingjiao represents Qingjiao chickens. (**d**) *METTL21C* mRNA expression level in toe elongation muscles. (**e**) *METTL21C* relative mRNA expression in caged and free-ranging chickens.* *p* ≤ 0.05; ** *p* ≤ 0.01.

**Table 1 genes-10-00628-t001:** Methyltransferase-like 21C (*METTL21C*) gene family information in chickens (*G. gallus*).

Gene Name	Protein ID	CDS ID	Gene ID	Chromosome	Gene Position	
					Start	End
*GgMETTL21C1*	*ENSGALP00000002756.3*	*ENSGALT00000002759.6*	*ENSGALG00000001790.6*	chr18	4,245,590	4,247,698
*GgMETTL21C2*	*ENSGALP00000003173.1*	*ENSGALT00000003178.2*	*ENSGALG00000002044.5*	chr14	13,793,041	13,797,471
*GgMETTL21C3*	*ENSGALP00000027192.5*	*ENSGALT00000027243.7*	*ENSGALG00000016864.7*	chr1	144,281,168	144,285,163
*GgMETTL21C4*	*ENSGALP00000027185.6*	*ENSGALT00000027236.7*	*ENSGALG00000016859.7*	chr1	144,171,934	144,190,015
*GgMETTL21C5*	*ENSGALP00000053406.2*	*ENSGALT00000048292.2*	*ENSGALG00000033603.3*	chr14	10,511,222	10,519,988
*GgMETTL21C6*	*ENSGALP00000047914.1*	*ENSGALT00000056245.2*	*ENSGALG00000035963.2*	chr8	5,470,226	5,471,907
*GgMETTL21C7*	*ENSGALP00000063245.1*	*ENSGALT00000086206.2*	*ENSGALG00000008493.7*	chr7	12,179,296	12,182,133
*GgMETTL21C8*	*ENSGALP00000032767.3*	*ENSGALT00000033408.5*	*ENSGALG00000008493.7*	chr7	12,179,725	12,182,133
*GgMETTL21C9*	*ENSGALP00000020014.7*	*ENSGALT00000020041.7*	*ENSGALG00000012269.7*	chr5	57,863,106	57,866,374
*GgMETTL21C10*	*ENSGALP00000073336.1*	*ENSGALT00000098371.1*	*ENSGALG00000051460.1*	chr5	57,862,387	57,869,575

Note: Protein ID, gene ID, CDS ID, and gene position indicate that the accession numbers of the *METTL21C* gene family member sequences were downloaded from the National Center for Biotechnology (NCBI). CDS: Coding sequence.

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
