# Peer review of "Genome-Wide Identification and Transcriptional Expression of the METTL21C Gene Family in Chicken"

_genes, 2019, doi:10.3390/genes10080628_

Round 1
Reviewer 1 Report
The authors described evolution of METTL21C, which might be important in chicken development. By looking at the presence of a particular motif, the author conclude conservative evolution of the protein. In addition, examining the expression pattern of the gene, the authors conclude this gene might be play an important role in the development of muscles.
The topic the authors treated is interesting, however, their conclusion is not convincing.
Introduction
1) is the location of 13q33.1 in the human genome?
2) from the introduction, METTL 21C seems to be a single copy gene. But later we know that the several copies of this gene dispersed in genome of humans, mice and chickens. This might be a cause of the confusion. If the multiple copy of this gene is a new finding by the authors, to describe clearly we can understand it.
3) line 57: domain D1 appears suddenly. It is better to describe a basic structure.
Result
1) line140-142: 119 genes detected were reduced to 53 genes. Causes for this reduction should be mentioned.
2) Figure 1: bootstrap probability should be added. This must be a key for grouping.
3) Figure 2: There are 10 chromosomes were drawn. However, in actual number of different chromosomes is six. The same chromosomes should be drawn as the same one.
4) Figure 2 legend: What does ”pseudo-chromosome” mean? Do you mean “a linkage group”. However, in chicken genome, I think chromosomal location of many genes has already determined.
5) In an expression experiment, what gene was used as “control”?
6) on page 8 line 9 from the bottom: why “Of course”?
7) page 9 the bottom line: what is your hypothesis?
Discussion
1) gene structure
the presence of a particular motif does not mean the conservativeness of the family.
Because sometimes we use the presence of a particular motif as the basis of definition of the “family”., the presence of a motif and conservativeness is a kind of tortology.
2) In discussion general,
Related to the above comment, the definition of family and sub family are very ambitious.
3) the definition of group A, B, C is also ambiguous. Based on phylogenetic analysis, you have already defined group I to VI.
Author Response
Response to Reviewer 1 Comments
Thank you for your valuable comments. We have studied the valuable comments from you carefully, made a significant effort to make the work clearer, and tried our best to revise the manuscript. The point to point responds to the reviewer’s comments as following:
Introduction
Point 1: is the location of 13q33.1 in the human genome?
Response: Thank you for your comments. We have added the sentence in the INTRODUCTION section from Line 43 to 44 as followings:
“in human, METTL21C is located on 13q33.1 and is thus also called “C13orf39”[5].
Point 2: from the introduction, METTL 21C seems to be a single copy gene. But later we know that the several copies of this gene dispersed in genome of humans, mice and chickens. This might be a cause of the confusion. If the multiple copy of this gene is a new finding by the authors, to describe clearly we can understand it.
Response: Thank you for your comments. The previous study reported the METTL21C is a single copy gene, but in our study, we identified a total of 10 genes as METTL21C gene family.
Point 3: line 57: domain D1 appears suddenly. It is better to describe a basic structure.
Response: Thank you for your comments. We have added some sentence in the INTRODUTION section from Line 72 to 74 as followings:
“ As mentioned above, VCP including four domains, the domain D1 is one of the two AAA ATP-binding domains, So the mutation of VCP gene involved in the pathogenesis of IBMPFD[21]. ”
Result
Point 4: line140-142: 119 genes detected were reduced to 53 genes. Causes for this reduction should be mentioned.
Response: Thank you for your comments. We have added some sentences in the RESULT section from Line 179 to 181 as followings:
“To check for the presence of the lysine methyltransferase domain, those 119 candidate protein sequences are reduced to 53 genes which contain the same conserved domain is lysine methyltransferase.”
Point 5: Figure 1: bootstrap probability should be added. This must be a key for grouping.
Response: Thank you for your comments. We have remake Figure1 and add the bootstrap value.
Point 6: Figure 2: There are 10 chromosomes were drawn. However, in actual number of different chromosomes is six. The same chromosomes should be drawn as the same one.
Response: Thank you for your comments. We have remake Figure2 and also revised the RESULT section from Line 200 to 207 as following:
“ The 10 METTL21C genes were distributed unevenly in the G. gallus chromosomes based on the physical positions. We found the distribution of the METTL21C genes were different on each of 6 chromosome (Table 1; Figure 2). The accession number of the coding sequence and the genomic and protein IDs are listed in Table 1. Chromosomes 8 and 18 contained only one gene GgMETTL21C1 and GgMETTL21C6, whereas GgMETTL21C3 and GgMETTL21C4 were both found on chromosome 1 but in different positions. GgMETTL21C2 and GgMETTL21C5, GgMETTL21C7 and GgMETTL21C8 , as well as METTL21C9 and GgMETTL21C10 were also found at different positions of the same chromosomes, such as chr14, chr7 and chr5, respectively ( Figure 2 ). ”
Point 7: Figure 2 legend: What does ”pseudo-chromosome” mean? Do you mean “a linkage group”. However, in chicken genome, I think chromosomal location of many genes has already determined.
Response: Thank you for your comments. We have changed the word “pseudo-chromosome” to “chromosome”.
Point 8: In an expression experiment, what gene was used as “control”?
Response: Thank you for your comments.
We wrote the sentence in the RESULT section from Line 318 as following:“Using β-actin as the internal reference gene ”.
Point 9: on page 8 line 9 from the bottom: why “Of course”?
Response: Thank you for your comments.We have revised the words “Of course” to “based on the different expression level” in RESULTS section from Line 272 to 273.
Point 10: page 9 the bottom line: what is your hypothesis?
Response: Thank you for your comments. We have revised the word“hypothesis”to “which is consistent with our expression profile analysis of GgMETTL21C3 in ly chicken” from line 317 to 318.
Discussion
Point 11: gene structure
the presence of a particular motif does not mean the conservativeness of the family. Because sometimes we use the presence of a particular motif as the basis of definition of the “family”., the presence of a motif and conservativeness is a kind of tortology.
Response: Thank you for your comments. We have revised the sentence in the DISCUSSION section in Line 416.
“ In conclusion, based on these analyses, including motif and gene structure, showed that METTL21C genes were conservative in evolution.”
Point 12: In discussion general,
Related to the above comment, the definition of family and sub family are very ambitious.
Response: Thank you for your comments. We have added some sentences in the DISCUSSION section from Line 342 to 345 as followings:
“ The gene family is structurally and functionally similar and contains a common domain. For the METTL21C, the same domain is lysine methyltransferase domain determined a total of 10 memers grouping into METTL21C gene family. Based on phylogenetic analysis, the METTL21Cs are divided into six subfamilies was group I to VI. ”
Point 13: the definition of group A, B, C is also ambiguous. Based on phylogenetic analysis, you have already defined group I to VI.
Response: Thank you for your comments. We have added some sentences in the RESULTS section from Line 184 to186 as followings:
Based on the completed alignment of the sequences, and the phylogenetic analysis, we divided a total of 53 members of chicken, mice and human into six groups, named group I to VI.
We have added some sentences in the RESULTS section from Line 271 to272 as followings:
“Based on phylogenetic analysis and expression level, we divied the 10 members of chicken into three groups, named group A, B, and C. ”
We have revised sentences in the DISSCUSSION section from Line 444 to 445 as followings:
“Based on the phylogenetic analysis and METTL21Cs’ expression level, we divided these 10 genes into three groups. ”
Thank you for your consideration our manuscript of “Genome-Wide Identification and Transcriptional Expression of the METTL21C Gene Family in Chicken " publish on Journal of Genes.
Submission Date
26 June 2019
Date of this review
25 Jul 2019 12:30:21

Reviewer 2 Report
This is a well written manuscript that explores in depth the relative conservation and potential function of the METTL21C gene family in skeletal muscle development in chickens.
The following issues must be addressed prior to publication (given by line number):
29) ***Need to lead in with big picture concept of muscle/meat development and why it is important to study the gene family in question rather than leading with detailed information about specific genes of interest. Readers need to understand importance or reason for studying these genes in these species first, just like in your abstract.
114) Confused about timing of sampling, were 6 chickens of different breeds sampled at
each of the 3 timepoints or were there just 6 samples total? Need to describe which breeds were sampled at which timepoints and why not all breeds were sampled at all time points. Must also include information about the
husbandry of the birds prior to sample collection- how were birds
housed, fed and watered, seasonal conditions, light exposure
times/levels, etc. Were all birds raised free-range or were some
cage-raised? Please describe environmental conditions (temp, humidity, etc). This may
necessitate another paragraph or subsection within Methods. All of these
factors can significantly influence skeletal muscle development and gene expression
profiles.
128) Which skeletal muscle and which region of that
muscle was sampled?
Was the same anatomic site sampled for muscle, liver, and skin in each
bird? Our research has found
that gene expression can even vary between cranial and caudal aspects
of the developing pectoral (breast) muscle in young chickens.
Colleagues have found expression differences across liver lobes too.
Results section- Were there observable growth differences between the
birds of different breeds at each collection time point? If available,
please report bird weight differences and any measured difference in
muscle mass between the birds since you have linked your gene expression
to possible muscle mass development differences in the chickens.
Discussion section- Needs to be streamlined and refocused toward greater context of muscle development. Poor flow for reader in the first paragraph because ideas jump around from difference organ systems from osteoporosis (skeletal) to brain development. Would recommend instead leading in with the paragraph that begins "The METTL21C genes of chicken contained three conserved domains identified in NCBI-Batch-CDD, including..." instead to highlight the main findings of your study. As in introduction, you should lead in with the big picture context about why your work and comparison finding are important to our understanding of genetics and gene expression underlying muscle development before going into specifics on gene naming. The you can move the first paragraph later in the discussion since it focuses on less important renaming rather than your main findings.
Figure 4- If available, should also include expression profiles of Lfh and qjm at day 60 and 120 and sy and Ly at day 20. Difficult to make accurate comparison when different breeds and ages are put side by side without having all time points for all breeds. Cannot tell whether possible expression differences are due to breed differences, time of development, or just individual birds the way this is currently presented.
Figure 5.
Confused by a and b, since there was no mention of embryonic work in the
methods. Please add embryonic work to the Methods section. Also
confused by muscle source selected. Why were these particular muscles
selected for analysis? Please describe reason in methods. "Flatfish"
needs to be changed to "soleus" throughout the document since the term
flatfish refers to a type of fish, not a muscle group. Please add gene
name to part d y-axis title. Please add description of tissue source
for part e. Was this performed in skeletal muscle? Which one?
The following items are additional recommendations for minor revision to enhance clarity for readers (given by line number):
13 and 70) "more skeletal muscle mass" not "more skeletal muscles". Free range chickens may add muscle mass physiologically but do not have extra muscles. Need to also include comparator chicken type- "more muscle mass and higher egg production than ______ chickens" (what type are you using for comparison?) Is this finding true for all types of chickens, or just for layer type birds?
23/24) Change to "expression level increased in conjunction with enhanced? muscle development of chicken muscle." How did you clinically link gene expression with differences in bird growth and muscle development?
25) Only in chickens or across multiple animal species?
30) Need to indicate that this locus is for human chromosomes, since multiple species are studied but chromosomal location could vary across species.
38) genome wide studies in humans. (or one of the other animal species?)
55) Do Paget's and IBMFD affect all types of muscle (cardiac, smooth, etc.) or just skeletal? Just curious because you are building the case for these genes specifically aiding skeletal muscle development.
64) "Gallus gallus domesticus" is commonly used to differentiate from species name of wild junglefowl
67) Is black-bone chicken raised as layer (egg-type), broiler (meat-type) or dual purpose (egg and meat)?
69) Cage should be "caged" or "cage-raised."
71) In this paragraph, you need to indicate whether study only included Lueyang black-bone chicken sequences or also included other chicken breeds raised globally. This information could change the ability to extrapolate beyond the breed/s studied to chickens/poultry as a whole. Also need to indicate comparator species (mouse, human) in this paragraph since you are using these as "knowns" to study your unknown.
85) Please indicate chicken breed/s for the whole protein sequences used.
112) These are different breeds, not different species. What tissues were sampled for this analysis?
159) Since localization was accomplished to specific chromosomes, I would recommend slightly rephrasing these sentences to mention specific chromosome numbers in text so that the written information stands alone from the table and figure too. For example, "distribution of the ... genes was different on each of ____ chromosomes (how many total chromosomes?). Chromosomes #__________ and #___ contained one gene each, whereas ...C3 and ...C4 were both found on chromosome 1 but in different positions. ...C7 and 8 as well as ...9, and 10 were also found at different positions of the same chromosomes, chr7 and chr5, respectively.Line numbers disappeared after Pg5, so I have itemized suggested revisions by page number
Pg 7- What do you mean by "almost" in the phrases starting with "For example, Group Ⅰ almost had one exon"- Do you mean that "almost all" members of the group had one exon or that members of the group somehow did not have a complete exon?
Should state "Motif 1 was present in all METL21C genes except..." not "species"
Motif 2 "expect" > "except"
Pg10
"was highly expressed in free ranging compared with caged chickens" This finding would be expected physiologically given that you sampled leg muscles in ambulatory (free range) versus sedentary (caged) birds. Did you also sample other muscle groups, like pectoral or back musculature to see if this held true across muscle groups?
Who is "Marie" in the phrase "Marie also found that the methylation process involved in METTL23 affects the nervous system." Should use author's last name.
"GgMETTL21C1 also has the conserved domain of SAM, and Raical SAM proteins are involved in catalytic reactions, including aberrant methylation, isomerization, sulfur insertion, ring formation, anaerobic oxidation, and protein free-radical formation [47]. We believe that GgMETTL21C1 has a similar function in chicken." Do you mean that GgMETTL21C1 contributes to processes that damage cells or is it likely protective against these processes? As a heat shock protein, I would have guessed that it helped prevent these damaging effects on cells.
In line 144 you state, "G. gallus, M. musculus, and H. sapiens protein sequences was constructed. These three species had 10, 11, and 32 METTL21C genes, respectively." However, in the discussion, you state "The present study identified 10 METTL21C genes in chicken, while higher animals such as M. musculus and H. sapiens had fewer METTL21C genes in their genomes (Table 1)." These statements contradict one another since you have shown mice and humans to have more of the genes than chickens. Please correct or clarify.
Pg 11
First line of 4.3- Change species to breed
"In previous studies, the METTL21C genes showed different expression patterns during different periods (20, 60, and 120 days) in Lfh, Qjm, Sy, and Ly." Need to give citations for this statement.
"GgMETTL21C5 and GgMETTL21C10 were found to be highly expressed in chicken cells"- In which tissues?
Pg12
The 10 GgMETTL21Cs were respectively distributed in seven different chromosomes,[ADD: with some colocalized to the same chromosome] such as GgMETTL21C7 and GgMETTL21C8 located in chromosome7
"flatfish muscle, long extensor digitorum, skeletal muscle, myocardium, and leg muscle"- Should just read "soleus, long extensor digitorum, and myocardium", since skeletal muscle and leg muscle redundantly describe the specific muscles tested. Alternately could read "skeletal muscle of the leg and cardiac muscle" which would also cover all the specific muscles mentioned.
Figures and figure legends:
Figure 2- Is there anyway to combine genes found together on a chromosome onto the same line? This would better illustrate their physical relationship to one another.
Figure 3 legend- recommend also including the species abbreviation associated with the gene sequence origins after the species names "G. gallus (Gg), M. musculus (Mm), H. sapiens (Hs). Seems intuitive, but may not be readily apparent to interested readers outside the field of genetics, such as poultry producers and veterinarians.
Author Response
Response to Reviewer 2 Comments
Thank you for your valuable comments. We have studied the valuable comments from you carefully, made a significant effort to make the work clearer, and tried our best to revise the manuscript. The point to point responds to the reviewer’s comments as following:
Point 1: 29) ***Need to lead in with big picture concept of muscle/meat development and why it is important to study the gene family in question rather than leading with detailed information about specific genes of interest. Readers need to understand importance or reason for studying these genes in these species first, just like in your abstract.
Response: Thank you for your comments. We have added some sentences in the INTRODUCTION section from Line 29 to 39 as followings:
“The chicken (Gallus gallus domesticus) may originate from red junglefowl (G. gallus) [1]. Lueyang black-bone chicken, is a famous native chicken breed with some distinctive features, such as its low fat content, strong muscles, and considerable nutritional and medicinal value. It has been listed in the Poultry Genetic Resources of China and was identified as a National Geographical Indication Product [2]. Chicken feeding methods are generally divided into caged and free range. It has been found that free-range chickens have more muscle mass [3]. A previous study showed that METTL21C is a skeletal-muscle-specific lysine methyltransferase [4]. As we all know, skeletal muscle plays an important role in the development of chicken, and can affect the athletic ability of the chicken. Thus, it is relevant to explore the role of the METTL21C gene in the muscle development of chicken species.”
We have added some sentences in the INTRODUCTION section from Line 87 to 88 as followings:
“ The METTL21C gene family from G.gallus domesticus offers an exceptional opportunity to study the evolution and function of mucle development systems[24].”
Point 2: 114) Confused about timing of sampling, were 6 chickens of different breeds sampled at each of the 3 timepoints or were there just 6 samples total? Need to describe which breeds were sampled at which timepoints and why not all breeds were sampled at all time points. Must also include information about the husbandry of the birds prior to sample collection- how were birds housed, fed and watered, seasonal conditions, light exposure times/levels, etc. Were all birds raised free-range or were some cage-raised? Please describe environmental conditions (temp, humidity, etc). This may necessitate another paragraph or subsection within Methods. All of these factors can significantly influence skeletal muscle development and gene expression profiles.
Response: Thank you for your comments. We have added some sentences in the Materials and Methods section from Line 133 to 148 as followings:
“ A total of six different breeds of black-bone chickens, such as 20 days of Liangfenghua chicken, 20 days of Qingjiaoma chicken, 60 and 120 days of Siyu chicken, and 60 and 120 days of Lueyang black-bone chickens are collected. For each chicken development time, we considered periods of 20, 60, and 120 days. We raise these chickens in one way of cage and free-range ways, of which Lueyang black-bone is free-range and the rest are caged. Different breeds of chickens are different feeding conditions. Under normal circumstances, the optimal temperature for each stage of silky fowl is: brooding period (0~6 weeks old), 0~3 days old is 38~36°C, and then decreased by 2~3°C every week to 6 weeks old. The temperature should be gradually reduced by 21~18 °C; the optimum temperature for broiler chicken after 6 weeks of age is 21~16 °C. The humidity in the house should be controlled by a hygrometer. The optimum suitable humidity for chicks is 55%~65%, and that of adult chickens is 60%~70%. Regarding the illumination, chicks of 1 to 3 days old maintain 23 to 24 hours of light, 4 to 14 days of age maintain 16 to 19 hours of light, and after 15 days of age, 8 to 9 hours of light are maintained, and artificial light is applied when sunshine is insufficient. Free-range care generally does not require artificial control of the growing environment, and if necessary, some measures are taken according to the weather.”
Point 3: 128) Which skeletal muscle and which region of that muscle was sampled? Was the same anatomic site sampled for muscle, liver, and skin in each bird? Our research has found that gene expression can even vary between cranial and caudal aspects of the developing pectoral (breast) muscle in young chickens. Colleagues have found expression differences across liver lobes too.
Response: Thank you for your comments. We have added some sentences in the Materials and Methods section from Line 164 to 167 as following:
“Because METTL21C is related to muscle development, so we collect the skeletal muscle of different breeds at the same anatomic site, such as Siyu, Qingjiaoma, Lueyang, and Liangfenghua, collecting the soleus muscle of the Lueyang chicken and Qingjiaoma and stomach, gizzard, lung, skeletal muscle, liver, skin, myocardium from Ly chicken(Table S1, Supplementary Materials). ”
Point 4: Results section- Were there observable growth differences between the birds of different breeds at each collection time point? If available, please report bird weight differences and any measured difference in muscle mass between the birds since you have linked your gene expression to possible muscle mass development differences in the chickens.
Response: Thank you for your comments. We have added some sentences in the RESULTS section from Line 314 to 315 as followings:
“ These different breeds of chickens do not have significantly for different growth differences in morphology.”
Point 5: Discussion section- Needs to be streamlined and refocused toward greater context of muscle development. Poor flow for reader in the first paragraph because ideas jump around from difference organ systems from osteoporosis (skeletal) to brain development. Would recommend instead leading in with the paragraph that begins "The METTL21C genes of chicken contained three conserved domains identified in NCBI-Batch-CDD, including..." instead to highlight the main findings of your study. As in introduction, you should lead in with the big picture context about why your work and comparison finding are important to our understanding of genetics and gene expression underlying muscle development before going into specifics on gene naming. The you can move the first paragraph later in the discussion since it focuses on less important renaming rather than your main findings.
Response: Thank you for your comments. We revised the sentences in the DISCUSSION section from Line 330 to 369 as followings:
“ METTL21C is a skeletal-muscle-specific lysine methyltransferase which plays a critical role in the development of bones and muscles [4]. The METTL21C gene family is closely related to osteoporosis and sarcopenia by the NF‐kB signaling pathway to prevent the differentiation of myoblasts [11]. The METTL21C genes of chicken contained three conserved domains identified in NCBI-Batch-CDD, including the AdoMet_MTases superfamily, SAM_superfamily, and FAM86, suggesting that METTL21Cs are extremely conservative in evolution(Figuer S1). We found all chicken METTL21Cs had the conserved domain of AdoMet_MTases. The key domain is AdoMet_MTases (Figure S1), and in general, the most common methyl donor is S-adenosyl-L-methionine (AdoMet). Methylation reactions often occur in cells, which are catalyzed by MTases and induce methyl transfer from donor substrate molecules, implying that these genes have a similar function (Figure S1) [43]. In chicken, the number of domains of AdoMet_MTases is larger than that of other domains, such as the domain of SAM and FAM86.
The gene family is structurally and functionally similar and contains a common domain. For the METTL21C, the same domain is lysine methyltransferase domain determined a total of 10 memers grouping into METTL21C gene family. Based on phylogenetic analysis, the METTL21Cs are divided into six subfamilies was group I to VI. The present study identified 10 METTL21C genes in chicken, while less than M. musculus and H. sapiens (Table 1). The proteomes of these animals are different. For example, the M. musculus proteome is about 46,930 kb, and the H. sapiens proteome is about 70,940 kb. However, the G. gallus proteome is about 25,209 kb. Comparative analysis suggested that chickens, mice, and humans have similar numbers of gene families [44-46], indicating that the number of METTL21C in different species is not determined by the genome size (Table 1).
In previous studies, GgMETTLC1 was called METTL23, which is a GABPA (GA-binding protein transcription factor, alpha subunit) function regulator which plays an important role in the transcriptional regulation of human cognition [47-48]. and affects the nervous system and brain development [49]. Thus, we speculated that GgMETTL21C1 could be related to brain development in chickens. In terms of GgMETTL21C2, containing FAM86A protein belongs to the protein MTases family which catalyzes the trimethylation of eukaryotic elongation factor 2 (eEF2) on Lys-525. [50]. indicating that the GgMETTL21C2 is involved in protein synthesis, which is critical for cell survival during stress [51].. In this study, it highly expressd in Lueyang chicken, compared to other breeds (Figure 4 and 5), and highly expressed in skeletal muscle compaered to other muscle (Figure 5e). With the development of muscle of chicken, the expression level of GgMETTL21C2 is increased, these results show that GgMETTL21C2 is related to the development of skemetal muscle of Lueyang breeds.(Figure 4 and 5 a, b, c and d)In a previous study, Philippe referred to GgMETTL21C5 as METTL22, which can mediate methylation of kin17 on lysine 135. kin17 is a DNA/RNA-binding protein that participates in the replication and repair of DNA and the process of pre-mRNA [52], so we found that GgMETTL21C5 plays an important role in chickens. In a previous study, Magnus found that METTL21A acts as a protein MTase, which is involved in the trimethylation of a conserved lysiner This study also named the unnamed ID, which will enable others to identify the gene more clearly (Table S5, Supplementary Materials).”
Point 6: Figure 4- If available, should also include expression profiles of Lfh and qjm at day 60 and 120 and sy and Ly at day 20. Difficult to make accurate comparison when different breeds and ages are put side by side without having all time points for all breeds. Cannot tell whether possible expression differences are due to breed differences, time of development, or just individual birds the way this is currently presented.
Response: Thank you for your comments. In our study, we only sequence Lfh and Qjm at day 20 and sy and Ly at day 60 and 120.
Point 7: Figure 5. Confused by a and b, since there was no mention of embryonic work in the methods. Please add embryonic work to the Methods section. Also confused by muscle source selected. Why were these particular muscles selected for analysis? Please describe reason in methods. "Flatfish" needs to be changed to "soleus" throughout the document since the term flatfish refers to a type of fish, not a muscle group. Please add gene name to part d y-axis title. Please add description of tissue source for part e. Was this performed in skeletal muscle? Which one?
Response: Thank you for your comments. We revised the word "Flatfish" to "soleus". We have add gene name to part d y-axis title. About the figure 4 a and b, we have added the methods from line 164 to 167 as following:
“ Because METTL21C is related to muscle development, so we collect the skeletal muscle of different breeds at the same anatomic site, such as Siyu, Qingjiaoma, Lueyang, and Liangfenghua, collecting the soleus muscle of the Lueyang chicken and Qingjiaoma and stomach, gizzard, lung, skeletal muscle, liver, skin, myocardium from Ly chicken(Table S1, Supplementary Materials). ”
Minor concerns
Point 8: 13 and 70) "more skeletal muscle mass" not "more skeletal muscles". Free range chickens may add muscle mass physiologically but do not have extra muscles. Need to also include comparator chicken type- "more muscle mass and higher egg production than ______ chickens" (what type are you using for comparison?) Is this finding true for all types of chickens, or just for layer type birds?
Response: Thank you for your comments. We have revised "more skeletal muscle mass" to "more skeletal muscles". We compared with free rang chicken and ranged chicken to analysed muscles mass. We have deleted the sentence “higher egg production”from Line 13.
Point 9: 23/24) Change to "expression level increased in conjunction with enhanced? muscle development of chicken muscle." How did you clinically link gene expression with differences in bird growth and muscle development?
Response: Thank you for your comments. As shown in (Figure5, c and d), with the ages increased of muscle, the expression level of METTL21C3 is gradually increased, so we claimed that the clinically link gene expression with differences in chicken growth and muscle development.
Point 10: 25) Only in chickens or across multiple animal species?
Response: Thank you for your comments. In our research, we fed chickens in two ways, one in cages and the other in stocking. During the development of growth in chicken, we gather the sample of muscle. We only use chicken in our study.
Point 11: 30) Need to indicate that this locus is for human chromosomes, since multiple species are studied but chromosomal location could vary across species.
Response: Thank you for your comments. We have added the sentence from line 43 as following: “In human, METTL21C”
Point 12: 38) genome wide studies in humans. (or one of the other animal species?)
Response: Thank you for your comments. We have added the sentence from line 51 as following:
“a bivariate genome-wide association studies (GWAS) have found that METTL21C can also act as a pleiotropic factor on muscles [11].”
Point 13: 55) Do Paget's and IBMFD affect all types of muscle (cardiac, smooth, etc.) or just skeletal? Just curious because you are building the case for these genes specifically aiding skeletal muscle development.
Response: Thank you for your comments. In our study, we only pay attention to the skeletal in Paget's and IBMFD, so we don’t know METTL21C could influence the other muscle.
Point 14: 64) "Gallus gallus domesticus" is commonly used to differentiate from species name of wild junglefowl
Response: Thank you for your comments. We have revised "Gallus gallus" to " Gallus gallus domesticus ".
Point 15: 67) Is black-bone chicken raised as layer (egg-type), broiler (meat-type) or dual purpose (egg and meat)?
Response: Thank you for your comments. The black-bone chicken raised as dual purpose ( egg and meat ), but it has low fat content, strong muscles, and considerable nutritional and medicinal value. so, we choose this chicken in our study.
Point 16: 69) Cage should be "caged" or "cage-raised."
Response: Thank you for your comments. We have revised word "cage" to "caged".
Point 17: 71) In this paragraph, you need to indicate whether study only included Lueyang black-bone chicken sequences or also included other chicken breeds raised globally. This information could change the ability to extrapolate beyond the breed/s studied to chickens/poultry as a whole. Also need to indicate comparator species (mouse, human) in this paragraph since you are using these as "knowns" to study your unknown.
Response: Thank you for your comments. We have added the sentence from line 88 to 94 as following:
“In this study, to detect the mechanism of METTL21C in the development of chicken muscles, According the global Genomic data on chickens, mouse and human published by others, we systematically characterized 10 METTL21C genes of G. gallus.. 11 METTL21C genes of M. musculus, and 32 METTL21C genes of H. sapiens. We used an iterative process of manual and computational analysis to identify members of G. gallus, M. musculus and H. sapiens METTL21C encoding a gene family within the latest-released G. gallus, M. musculus and H. sapiens whole-genome sequence [25-27].”
Point 18: 85) Please indicate chicken breed/s for the whole protein sequences used.
Response: Thank you for your comments. We have added the chicken for the whole protein sequences used on line 106.
Point 19: 112) These are different breeds, not different species. What tissues were sampled for this analysis?
Response: Thank you for your comments. We have revised word "species" to "breed". We also added the sentence from line 164 to 167 as following:
“ Because METTL21C is related to muscle development, so we collect the skeletal muscle of different breeds at the same anatomic site, such as Siyu, Qingjiaoma, Lueyang, and Liangfenghua, collecting the soleus muscle of the Lueyang chicken and Qingjiaoma and stomach, gizzard, lung, skeletal muscle, liver, skin, myocardium from Ly chicken(Table S1, Supplementary Materials).”
Point 20: 159) Since localization was accomplished to specific chromosomes, I would recommend slightly rephrasing these sentences to mention specific chromosome numbers in text so that the written information stands alone from the table and figure too. For example, "distribution of the ... genes was different on each of ____ chromosomes (how many total chromosomes?). Chromosomes #__________ and #___ contained one gene each, whereas ...C3 and ...C4 were both found on chromosome 1 but in different positions. ...C7 and 8 as well as ...9, and 10 were also found at different positions of the same chromosomes, chr7 and chr5, respectively.Line numbers disappeared after Pg5, so I have itemized suggested revisions by page number
Response: Thank you for your comments. We have added some sentences as in the RESULTS section from Line 200 to 207 as followings:
“ The 10 METTL21C genes were distributed unevenly in the G. gallus chromosomes based on the physical positions. We found the distribution of the METTL21C genes were different on each of 6 chromosome (Table 1; Figure 2). The accession number of the coding sequence and the genomic and protein IDs are listed in Table 1. Chromosomes 8 and 18 contained only one gene GgMETTL21C1 and GgMETTL21C6, whereas GgMETTL21C3 and GgMETTL21C4 were both found on chromosome 1 but in different positions. GgMETTL21C2 and GgMETTL21C5, GgMETTL21C7 and GgMETTL21C8 , as well as METTL21C9 and GgMETTL21C10 were also found at different positions of the same chromosomes, such as chr14, chr7 and chr5, respectively ( Figure 2 ).”
Point 21: Pg 7- What do you mean by "almost" in the phrases starting with "For example, Group Ⅰ almost had one exon"- Do you mean that "almost all" members of the group had one exon or that members of the group somehow did not have a complete exon?
Response: Thank you for your comments. We have added the sentence from line 239 to 241 as following:
“ For example, GroupⅠmost had one exon, such as GgMETTL21C6, HsMETTL21C1, HsMETTL21C2, MmMETTL21C3 and MmMETTL21C4 has one exon, but HsMETTL21C23 has 6 exon, GgMETTL21C10 has 5 exon. ”
Point 22:Should state "Motif 1 was present in all METL21C genes except..." not "species" Motif 2 "expect" > "except"
Response: Thank you for your comments. We have revised "species" to "breeds" in Line 249, and "expect" to "except" in Line 250.
Point 23: Pg10"was highly expressed in free ranging compared with caged chickens" This finding would be expected physiologically given that you sampled leg muscles in ambulatory (free range) versus sedentary (caged) birds. Did you also sample other muscle groups, like pectoral or back musculature to see if this held true across muscle groups?
Response: Thank you for your comments. Here, we conclude that the METTL21C gene is highly expressed in muscle-rich areas, demonstrating that this gene is involved in muscle production and is not intended to demonstrate that the gene is associated with a certain type of muscle. So, we compared with caged and free ranging chicken.
Point 24: Who is "Marie" in the phrase "Marie also found that the methylation process involved in METTL23 affects the nervous system." Should use author's last name.
Response: Thank you for your comments. We have deleted this sentence.
Point 25: "GgMETTL21C1 also has the conserved domain of SAM, and Raical SAM proteins are involved in catalytic reactions, including aberrant methylation, isomerization, sulfur insertion, ring formation, anaerobic oxidation, and protein free-radical formation [47]. We believe that GgMETTL21C1 has a similar function in chicken." Do you mean that GgMETTL21C1 contributes to processes that damage cells or is it likely protective against these processes? As a heat shock protein, I would have guessed that it helped prevent these damaging effects on cells.
Response: Thank you for your comments. We have deleted this sentence.
Point 26: In line 144 you state, "G. gallus, M. musculus, and H. sapiens protein sequences was constructed. These three species had 10, 11, and 32 METTL21C genes, respectively." However, in the discussion, you state "The present study identified 10 METTL21C genes in chicken, while higher animals such as M. musculus and H. sapiens had fewer METTL21C genes in their genomes (Table 1)." These statements contradict one another since you have shown mice and humans to have more of the genes than chickens. Please correct or clarify.
Response: Thank you for your comments.
We revised the word: "fewer" to "more" in Line 346.
Point 27: Pg 11
First line of 4.3- Change species to breed
"In previous studies, the METTL21C genes showed different expression patterns during different periods (20, 60, and 120 days) in Lfh, Qjm, Sy, and Ly." Need to give citations for this statement.
"GgMETTL21C5 and GgMETTL21C10 were found to be highly expressed in chicken cells"- In which tissues?
Response: Thank you for your comments. We have revised "species" to " breeds" and added the citations for this statements followings in Line 449.
We also deleted sentence in the DISSCUSSION section from Line 456-458 as followings:
“ GgMETTL21C5 and GgMETTL21C10 were found to be highly expressed in chicken cells, which is consistent with a previous study which suggested that the MTTTL21C gene family participates in myocyte differentiation and can promote the differentiation of myoblasts to myotubes [6]. ”
Point 28: Pg12
The 10 GgMETTL21Cs were respectively distributed in seven different chromosomes,[ADD: with some colocalized to the same chromosome] such as GgMETTL21C7 and GgMETTL21C8 located in chromosome7
Response: Thank you for your comments. We have added the word in the CONCLUSION section from Line 484 to 485 as followings:
“ with some colocalized to the same chromosome ”.
Point 29: "flatfish muscle, long extensor digitorum, skeletal muscle, myocardium, and leg muscle"- Should just read "soleus, long extensor digitorum, and myocardium", since skeletal muscle and leg muscle redundantly describe the specific muscles tested. Alternately could read "skeletal muscle of the leg and cardiac muscle" which would also cover all the specific muscles mentioned.
Response: Thank you for your comments. We have revised the sentence "flatfish muscle, long extensor digitorum, skeletal muscle, myocardium, and leg muscle" to "soleus, long extensor digitorum, and myocardium" in line 495.
Point 30: Figure 2- Is there anyway to combine genes found together on a chromosome onto the same line? This would better illustrate their physical relationship to one another.
Response: Thank you for your comments. We have revised the Figure 2.
Point 31: Figure 3 legend- recommend also including the species abbreviation associated with the gene sequence origins after the species names "G. gallus (Gg), M. musculus (Mm), H. sapiens (Hs). Seems intuitive, but may not be readily apparent to interested readers outside the field of genetics, such as poultry producers and veterinarians.
Response: Thank you for your comments. We have added species abbreviation associated with the gene sequence origins after the species names on Figure 3 legend-recommend in Line 228.
Thank you for your consideration our manuscript of “Genome-Wide Identification and Transcriptional Expression of the METTL21C Gene Family in Chicken" publish on Journal of Genes.
Submission Date
26 June 2019
Date of this review
24 Jul 2019 12:39:32

Round 2
Reviewer 1 Report
The authors revised well and answered my comments. Th revision looks fine.
This manuscript is a resubmission of an earlier submission. The following is a list of the peer review reports and author responses from that submission.